

# The classical Heisenberg model on the centred pyrochlore lattice

Rajah P. Nutakki[1,2]*, Ludovic D. C. Jaubert[3] and Lode Pollet[1,2]

**1** Arnold Sommerfeld Center for Theoretical Physics, University of Munich,
Theresienstr. 37, 80333 Munich, Germany
**2** Munich Center for Quantum Science and Technology (MCQST),
Schellingstr. 4, 80799 Munich, Germany
**3** CNRS, Université de Bordeaux, LOMA, UMR 5798, 33400 Talence, France

★ Rajah.Nutakki@lmu.de

## Abstract

The centred pyrochlore lattice is a novel geometrically frustrated lattice, realized in the metal-organic framework $Mn(ta)_2$ [1] where the basic unit of spins is a five site centred tetrahedron. Here, we present an in-depth theoretical study of the $J_1 - J_2$ classical Heisenberg model on this lattice, using a combination of mean-field analytical methods and Monte Carlo simulations. We find a rich phase diagram with low temperature states exhibiting ferrimagnetic order, partial ordering, and a highly degenerate spin liquid with distinct regimes of low temperature correlations. We discuss in detail how the regime displaying broadened pinch points in its spin structure factor is consistent with an effective description in terms of a fluid of interacting charges. We also show how this picture holds in two dimensions on the analogous centred kagome lattice and elucidate the connection to the physics of thin films in $(d + 1)$ dimensions. Furthermore, we show that a Coulomb phase can be stabilized on the centred pyrochlore lattice by the addition of further neighbour couplings. This demonstrates the centred pyrochlore lattice is an experimentally relevant geometry which naturally hosts emergent gauge fields in the presence of charges at low energies.



# 1   Introduction

The study of frustrated magnetic systems [2] occupies an important position in modern condensed matter physics as a route to realizing states of matter exhibiting fractionalization, topological order and the emergence of gauge fields [3–5]. Such features can already emerge in classical systems, most famously in spin ice [6–8] where low-lying excitations may be described as magnetic monopoles interacting via an energetic Coulomb potential and entropic emergent gauge field. A similar picture extends to other spin models on the pyrochlore, such as the classical Heisenberg model, where the excitations are not monopoles of a true magnetic field, rather scalar charges of the emergent gauge field. This is known as a Coulomb phase [9, 10], since the low-energy theory above the vacuum ground state is classical electrostatics with charges interacting via effective Coulomb interactions. The appearance of such a phase is readily diagnosed by pinch point singularities in the spin structure factor and corresponding algebraic $1/r^3$ correlations in real space. To stabilize monopoles in ground states (of spin-ice systems) requires the use of magnetic fields [11–13], further neighbour exchange [14, 15], artificial interactions [16–18] or magneto-elastic coupling [19, 20], resulting in a monopole fluid, or, long-range order leading to the phenomenon of magnetic fragmentation [21].

    In the quantum case, although the ground state of the spin 1/2 Heisenberg model remains ambiguous, see e.g [22,23] and references therein, one can realize a $U(1)$ quantum spin liquid, effectively described by (compact) quantum electrodynamics, in the spin 1/2 XXZ model close to the Ising limit [24–28]. Here, the topological character of the ground state manifold of the Ising model on the pyrochlore is supplemented by quantum fluctuations to stabilize a massive superposition of topologically ordered states. Since the effective theory of the quantum spin liquid is in $(3 + 1)$ dimensions, the algebraic correlations go instead as $1/r^4$, destroying the sharp pinch points in the structure factor [29].

Recent work [1] has established that the metal-organic framework Mn(ta)$_2$ realizes a centred pyrochlore lattice, where the basic unit of spins is a five site centred tetrahedron. Comparison of bulk thermodynamic measurements to MC simulations suggest that Mn(ta)$_2$ is well approximated above $\sim 1$ K by a classical $J_1 - J_2$ Heisenberg model on the centred pyrochlore lattice, although ultimately dipolar interactions lead to ordering at lower temperatures. This opens up new avenues to explore frustrated magnetism beyond the pyrochlore lattice. In particular, the highly versatile nature of metal-organic frameworks [30] raises the possibility of engineering desired quantum or classical Hamiltonians on the centred pyrochlore lattice.

In this work, we perform a detailed theoretical study of the $J_1 - J_2$ classical Heisenberg model on the centred pyrochlore lattice, finding a rich phase diagram with competition between ferrimagnetic order on the one hand, and Coulomb physics on the other. This gives rise to unusual low temperature states of matter. Furthermore, this introduces a new paradigm of geometrically frustrated lattices based on centred units of spins where vertex sites are shared between adjacent clusters but central sites are not. Where a nearest neighbour spin model on the lattice made up of vertex sites can realize a Coulomb phase ground state, the addition of central spins introduces effective charges, exponentially screening spin correlations and causing the pinch points to acquire a finite width, as discussed in ref. [1]. In this paper we elaborate on this point, also demonstrating a similar effect on the 2D centred kagome lattice and making a connection to the physics of pyrochlore thin films, seen by mapping the periodic lattice in $d$-dimensional space to a $d + 1$-dimensional 'slab' with open boundaries in the additional dimension.

This article is organized as follows. In section 2 we introduce the lattice and model. Section 3 provides a brief summary of the main results. We then discuss our results for the $J_1 - J_2$ model, describing the ground state properties from an analytic perspective in section 4, the phase diagram obtained from Monte Carlo (MC) simulations in section 5 and finally describe the spin liquid in more detail in section 6, including discussion of the appropriate low-energy theory. In section 7 we present results for the analogous model on the centred kagome lattice, the 2D analogue of the centred pyrochlore lattice, before discussing the effect of an additional $J_3$ term on the centred pyrochlore in section 8. We conclude in section 9 with a summary and outlook.

## 2 Lattice and model

The centred pyrochlore lattice is obtained from the pyrochlore lattice [31] by the addition of a lattice site at the centre of each tetrahedron (see fig. 1a). Explicitly, it is defined by sites at positions

$$\mathbf{r}_{I,\mu} = \mathbf{R}_I + \boldsymbol{\delta}_\mu, \tag{1}$$

where $\mathbf{R}_I = n_1 \mathbf{a}_1 + n_2 \mathbf{a}_2 + n_3 \mathbf{a}_3$ are the sites of a face-centred cubic (fcc) lattice with integer $n_i$ and lattice vectors $\mathbf{a}_1 = \frac{1}{2}(1,1,0)$, $\mathbf{a}_2 = \frac{1}{2}(0,1,1)$, $\mathbf{a}_3 = \frac{1}{2}(1,0,1)$, and $\mu$ labels the six sublattices with basis vectors

$$\boldsymbol{\delta}_a = \mathbf{0}, \quad \boldsymbol{\delta}_b = \frac{1}{4}\begin{pmatrix} 1 \\ 1 \\ 1 \end{pmatrix}, \quad \boldsymbol{\delta}_1 = \frac{1}{8}\begin{pmatrix} 1 \\ 1 \\ 1 \end{pmatrix},$$

$$\boldsymbol{\delta}_2 = \frac{1}{8}\begin{pmatrix} -1 \\ -1 \\ 1 \end{pmatrix}, \quad \boldsymbol{\delta}_3 = \frac{1}{8}\begin{pmatrix} 1 \\ -1 \\ -1 \end{pmatrix}, \quad \boldsymbol{\delta}_4 = \frac{1}{8}\begin{pmatrix} -1 \\ 1 \\ -1 \end{pmatrix}. \tag{2}$$

All quantities are given in units where the side length of the conventional fcc unit cell $a = 1$. In what follows, we will refer to the sites at the centre of a tetrahedron, $\mu = a, b$, as central

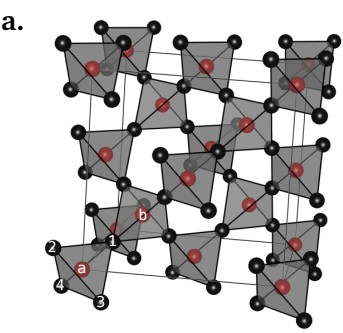

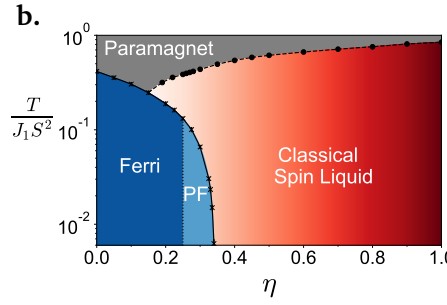

Figure 1: **a.** The conventional 24 site cubic unit cell of the centred pyrochlore lattice with the six basis sites labelled. **b.** Finite temperature phase diagram of eq. 3 for antiferromagnetic $J_1, J_2$ obtained from MC simulations for $L = 14$. Crosses are where there is a peak in the magnetic suceptibility, circles where $\partial L_t / \partial T$ is a maximum (see eq. 5). At $T = 0$, the ferrimagnetic (ferri) phase is characterized by saturated ferrimagnetic order, whereas the partial ferrimagnet (PF) remains unsaturated. No ordering is observed in the spin liquid regime for the temperatures simulated. The spin structure factor evolves continuously with $\eta$ in the spin liquid regime.

sites and those at the vertices of the tetrahedron, $\mu = 1, 2, 3, 4$, as vertex sites. The tetrahedra centred on $a(b)$ sites are referred to as $a(b)$ tetrahedra.

We consider the classical Heisenberg model on the centred pyrochlore lattice,

$$H = J_1 \sum_{\langle ij \rangle} \mathbf{S}_i \cdot \mathbf{S}_j + J_2 \sum_{\langle\langle ij \rangle\rangle} \mathbf{S}_i \cdot \mathbf{S}_j, \tag{3}$$

with exchange interactions of strength $J_1$ coupling nearest neighbours; the centre and vertex spins of a tetrahedron, and $J_2$ coupling next-nearest neighbours; the vertex spins on the same tetrahedron. In what follows we set $J_1 = S = 1$ and typically parametrize the model by $\eta = \frac{J_2}{J_1}$, using $\gamma = \frac{1}{\eta}$ instead when we would like to work close to the pyrochlore limit ($\eta = \infty$, where centre and vertex spins are decoupled).

## 3 Summary of results

The main result of this paper is the phase diagram presented in figure 1b. For $\eta \leq \frac{1}{4}$ the ground state is a ferrimagnet with all vertex and centre spins antiparallel. On the other hand, for $\eta > \frac{1}{4}$ the ground state is defined by a local constraint (eq. 6), where we find several unconventional low temperature states.

In the region $\frac{1}{4} < \eta < 0.343$, we find a partially ordered state with unsaturated ferrimagnetic order, retaining significant fluctuations in the magnetization. For $\eta > 0.343$ we find a disordered state characterized by distinct regimes of correlations (see figs 4d-f). For $\eta \lesssim 0.5$, at low $T$, the dominant features of the structure factor are diffuse, ferrimagnetic maxima which are indicative of short-ranged ferrimagnetic correlations. These correlations are not captured by mean-field calculations, indicating that the microscopic enforcement of the constraint on the lattice determines the correlation structure.

For $\eta \gtrsim 0.5$, the structure factor is characterized by broadened pinch points, which are well captured by our mean-field calculations. These show that the pinch points are never sharp for any finite $\eta$, so is not strictly a Coulomb phase. Instead the central spins act as fluctuating sources of flux, which leads to the broadening of the pinch points. Remarkably the width of the pinch points scales linearly with $\frac{1}{\eta}$ in the range $0.8 \lesssim \eta < \infty$ (see fig. 6), which can

be understood in terms of Debye screening in a charged fluid, where the charge strength is parameterized by $\eta$.

Similarly, we also compute the structure factor of the analogous $J_1 - J_2$ model on the 2D centred kagome lattice and also find broadened pinch points (fig. 8), providing evidence that this is a generic feature of lattices made up of centred corner-sharing units. Indeed, one can view the centred lattices as thin films of a higher dimensional lattice, which makes clear the connection between what we observe and previous examples of Coulomb phases destroyed by (reduced) lattice symmetry [32].

In addition, we show that by adding a small ferromagnetic $J_3$ one can stabilize a 3D Coulomb phase on the centred pyrochlore lattice (fig. 9), an example of how adding perturbations to the $J_1 - J_2$ Hamiltonian can pick out desired ground states. We also discuss the case of a large antiferromagnetic $J_3$ which leads to a state where Neel ordered centres and Coulomb phase vertex spins are entirely decoupled.

# 4 Ground state properties

## 4.1 Local constraint

For $J_2 < 0$ the model is unfrustrated and the ground state is a simple ferro or ferrimagnet, depending on the sign of $J_1$. In the ferrimagnet all central spins are anti-parallel to vertex spins. In this paper we focus on the (experimentally relevant [1]) quadrant of parameter space where $J_1 > 0, J_2 > 0$, which we call the centred pyrochlore Heisenberg antiferromagnet (CPHAF). We can map from $J_1$ to $-J_1$ by a global flip of all central spins, so the results presented here can be easily generalized to the $J_1 < 0$ region of the parameter space.

As for the pyrochlore Heisenberg antiferromagnet (PHAF) [33, 34], the Hamiltonian can be rewritten in terms of the tetrahedral units, $t$, of the lattice,

$$H = \frac{J_2}{2} \sum_t |\mathbf{L}_t|^2 - \frac{N}{3} \left( \frac{J_1^2}{2J_2} + 2J_2 \right), \tag{4}$$

however, due to the presence of the centre site, we require that $\mathbf{L}_t$ be given by

$$\mathbf{L}_t = \gamma \mathbf{S}_{t,c} + \sum_{v=1}^{4} \mathbf{S}_{t,v}, \tag{5}$$

where $\gamma$ rescales the contribution of the central spin. Centre sites are labelled by the index $c$, and the sum over $v$ runs over the vertices of the tetrahedron. The ground state is the state which minimizes $L_t = |\mathbf{L}_t|$ on all tetrahedra. For $\eta \leq \frac{1}{4}$, $L_t$ is minimized by the ferrimagnetic state, whereas for $\eta \geq \frac{1}{4}$, the ground state is defined by the local constraint

$$L_t = 0, \qquad \forall t. \tag{6}$$

On the pyrochlore lattice, such a constraint gives rise to an emergent $U(1)$ gauge field [9] and the subsequent Coulomb phase description [10].

## 4.2 Ising spins

To understand how the form of the Hamiltonian in equation 4 affects the possible ground states of the model, it is instructive to consider the analogous Ising model, where we replace



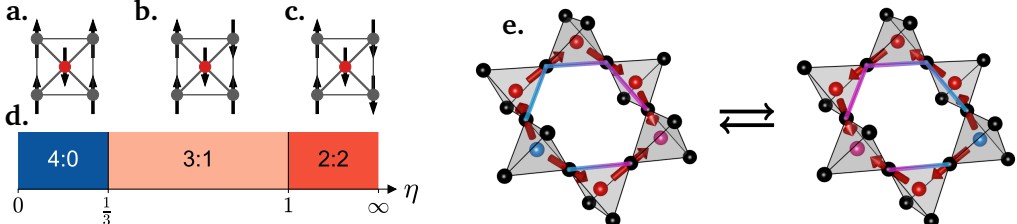

Figure 2: Ground states of the Ising model on the centred pyrochlore lattice. **a-c** Examples of the allowed single tetrahedron spin configurations in each of the ground states and **d.** the ground state phase diagram. For $\eta < 1/3$ (**a.**), the ground state is a long range ordered ferrimagnet. In the region $1/3 < \eta < 1$ (**b.**) the ground state manifold is made up of 3-up/1-down and 3-down/1-up vertex spin configurations with central spins correspondingly pointing opposite to the net polarization on each tetrahedron. For $\eta > 1$ (**c.**) the ground state is the spin ice state of the pyrochlore lattice, but with paramagnetic central spins which are decoupled from the vertex spins. **d.** Example of a move which changes the parity of the winding number in the $3:1$ ground state of the Ising model. Cyan (pink) spheres represent a centre site with spin $-1(+1)$ and we use the spin-ice convention for the spins at the vertices which are flipped during the move. All other spins residing on central (red) and vertex (black) sites remain unchanged. The move can be viewed as switching the direction of a pair of directed strings (highlighted) which begin and end on the same tetrahedra.

Heisenberg spins by Ising spins, $\mathbf{S}_i \to \sigma_i = \pm 1$. As for the Heisenberg Hamiltonian, the ground state is obtained by minimizing

$$L_t^I = \left| \gamma \sigma_{t,c} + \sum_{v=1}^{4} \sigma_{t,v} \right|, \qquad \forall t, \tag{7}$$

which gives the ground state phase diagram presented in figure 2d. Besides the ferrimagnetic ground state, there are also a pair of extensively degenerate disordered ground states. For $\eta > 1$ the ground state is the familiar spin-ice state of the antiferromagnetic Ising model on the pyrochlore lattice [6, 8]. The vertex spins of each tetrahedron must satisfy the 2-up/2-down $(2:2)$ rule, but now with an additional free spin variable occupying the central sites. This doubles the number of permutations of spin configurations allowed on a tetrahedron in the ground state to 12 and so by Pauling's argument [35] gives a residual entropy of $\ln(3)$ per tetrahedron.

In between the ferrimagnet and $2:2$ state, from $\frac{1}{3} < \eta < 1$, the ground state is where the vertex spins are either 3-up/1-down or 3-down/1-up configurations $(3:1)$ on *all* tetrahedra, with the corresponding central spins constrained to point antiparallel to the net moment of their vertex spins. There are 8 possible permutations of spin configurations giving a residual entropy of $\ln(2)$ per tetrahedron. Such $3:1$ single tetrahedra configurations have previously been studied in dilute concentrations in the context of excited states of spin ice, where the defects act as charges of the emergent gauge field [10]. In the presence of dipolar interactions in spin-ice, these charges become monopoles of a magnetic field. On the centred pyrochlore, the 'monopoles' (they are *not* sources of a physical magnetic field) are stabilized in the ground state of a large region of the parameter space, are disordered and have maximal density, with a monopole on each tetrahedron.

At, $\eta = 1$, the boundary of the 2 : 2 and 3 : 1 states, the ground state manifold contains any combinations of 2 : 2 and 3 : 1 states, with a large residual entropy of $\ln(5)$ per tetrahedron. Therefore the ground state manifold contains densities of monopoles from 0 to $N_t$, where $N_t$ is the number of tetrahedra, albeit at a fine-tuned point in the parameter space.

The 2 : 2 and 3 : 1 ground states can be distinguished by the topological nature of the respective ground state manifolds, characterized by a winding number or its parity respectively. For the 2 : 2 states, the central spins are entirely decoupled from the vertex spins so the $U(1)$ topological order of the spin ice ground state [24] is preserved. The connection to $U(1)$ topological order can be seen by mapping the vertex Ising spin variables $\sigma = +1(-1)$ to the presence (absence) of a dimer on the links of the diamond lattice. Any local operation (not encircling the system) which maintains the ground state condition will leave the number of dimers, $w_k$, crossing the plane perpendicular to $\hat{\mathbf{k}}$, invariant. This allows one to define the $U(1)$ winding numbers, $\mathbf{w} = (w_x, w_y, w_z)$, which label distinct topological sectors. However, in the 3 : 1 ground state, only the parity of these winding numbers are conserved by local operations, so one can instead define $\mathbb{Z}_2$ topological invariants. An example of a local operation which changes the winding number is presented in fig. 2e. In general, any pair of strings of vertex spins which begin and end on the same tetrahedra are now flippable, by flipping both of the centre spins at the beginning and end tetrahedra and all vertex spins in between. This is easiest to see in the spin-ice representation, where $\sigma_i = +1(-1)$ corresponds to a directed link variable pointing from $a$ to $b$ ($b$ to $a$ tetrahedra).

Therefore, the Ising model hosts distinct classical topological spin liquids at zero temperature, as seen for example in ref. [32] in spin ice thin films. As we discuss in section 6.2.2 there is also a more explicit connection to such thin films as a consequence of the geometry of the centred pyrochlore lattice. In the case of spin ice thin films, the transition between topologically ordered spin liquids requires a change in sign of the orphan bonds, whereas here this transition can occur by tuning the ratio of (antiferromagnetic) exchange couplings.

## 4.3 Degeneracy and flat bands

Returning to the Heisenberg model, we first consider how the form of the constraint (eq. 6) restricts the possible ground states of the model. For a single tetrahedron, the degree of ferrimagnetic correlation decreases continuously as $\eta$ is increased, from a saturated ferrimagnet at $\eta \le \frac{1}{4}$ to decoupled centre and vertex spins as $\eta \to \infty$. The corresponding Ising states form part of the Heisenberg ground state manifold at $\eta \le \frac{1}{4}, \eta = \frac{1}{2}$ and $\eta \to \infty$. The degeneracy, $D$, of the ground state manifold may be estimated for $\eta \approx 1$ using the counting argument of refs [33, 36, 37], yielding $D = 3N_t$ [1]. This is a higher degeneracy than the PHAF ground state, where $D = N_t$, with the additional degeneracy arising from the additional degrees of freedom carried by the (fixed length) central spin. Furthermore, the ground state degeneracy of a spin liquid can manifest itself in momentum space as degenerate flat bands, for example in the kagome [38] and pyrochlore ( [39]) antiferromagnets, with 1 out of 3 and 2 out of 4 flat bands respectively.

Here, both the generalized Luttinger-Tisza method (sec. 4.3.1) and the rewriting of the Hamiltonian in terms of a connectivity matrix (sec. 4.3.2) show that the disordered state of the CPHAF is characterized by a ground state with 4 out of 6 flat bands. As a result, the disordered ground state provides a large manifold of states to which perturbations could be added in order to stabilize particular ground states. For example, in section 8 we show how a 3D Coulomb phase can be stabilized by the addition of a small ferromagnetic $J_3$. Furthermore, this large degeneracy means that at finite temperature entropy can wash out the effect of small perturbations, maintaining the behaviour of the unperturbed $J_1 - J_2$ model, as demonstrated in ref. [1] in the case of dipolar interactions.

### 4.3.1 Luttinger-Tisza method

The generalized Luttinger-Tisza (LT) method [40, 41] is a mean-field method for obtaining the energy spectrum of a classical spin Hamiltonian in momentum space. To apply the LT, we first rewrite the Hamiltonian in Fourier space by introducing the momentum space spin variables

$$\mathbf{S}_{\mathbf{q}}^{\mu} = \frac{1}{\sqrt{N_{\text{u.c}}}} \sum_I e^{-i\mathbf{q}\cdot(\mathbf{R}_I + \delta_\mu)} \mathbf{S}_I^{\mu}, \tag{8}$$

where $I$ labels the primitive unit cell, $\mu$ the sublattice of the spin and $N_{\text{u.c}}$ is the number of primitive unit cells. This yields the Hamiltonian

$$\frac{H}{J_1} = \sum_{\mathbf{q}} \sum_{\mu,\nu} K_{\mathbf{q}}^{\mu\nu} \mathbf{S}_{\mathbf{q}}^{\mu} \cdot \mathbf{S}_{-\mathbf{q}}^{\nu}, \tag{9}$$

with (Hermitian) coupling matrix

$$K_{\mathbf{q}}^{\mu\nu}(\eta) = \begin{pmatrix} 0 & 0 & a_1 & a_2 & a_3 & a_4 \\ 0 & 0 & a_1^* & a_2^* & a_3^* & a_4^* \\ a_1^* & a_1 & 0 & c_{12} & c_{13} & c_{14} \\ a_2^* & a_2 & c_{12} & 0 & c_{23} & c_{24} \\ a_3^* & a_3 & c_{13} & c_{23} & 0 & c_{34} \\ a_4^* & a_4 & c_{14} & c_{24} & c_{34} & 0 \end{pmatrix}, \tag{10}$$

and components

$$a_\mu = \frac{1}{2} e^{-i\mathbf{q}\cdot\delta_\mu}, \tag{11}$$

$$c_{\mu\nu} = \eta \cos \mathbf{q} \cdot (\delta_\mu - \delta_\nu). \tag{12}$$

In the standard LT method [40], the strong constraint, that the spin on each lattice site is normalized,

$$|\mathbf{S}_i|^2 = 1, \qquad \forall i, \tag{13}$$

is replaced by the weak constraint,

$$\sum_i |\mathbf{S}_i|^2 = \sum_{\mathbf{q}} \sum_{\mu} \mathbf{S}_{\mathbf{q}}^{\mu} \cdot \mathbf{S}_{-\mathbf{q}}^{\mu} = N, \tag{14}$$

where the normalization is enforced only on average. Diagonalizing $K_{\mathbf{q}}^{\mu\nu}$, one can propose a ground state of the system by putting all of the weight from equation 14 into the mode at momentum $\mathbf{q}$ which corresponds to the minimum eigenvalue. However, this state will only be a valid physical ground state of the system if it also respects equation 13. In models with inequivalent spins, as is the case here, this standard method often fails to find physical states.

Lyons and Kaplan realized [41] that this can be remedied by modifying equation 14 to the form

$$\sum_I \sum_{\mu} \frac{|\mathbf{S}_I^{\mu}|^2}{\beta_\mu^2} = \sum_{\mathbf{q}} \sum_{\mu} \mathbf{t}_{\mathbf{q}}^{\mu} \cdot \mathbf{t}_{-\mathbf{q}}^{\mu} = N_{\text{u.c}} \sum_{\mu} \frac{1}{\beta_\mu^2}, \tag{15}$$

where we introduced the rescaled momentum space spin variables, $\mathbf{t}_{\mathbf{q}}^{\mu} = \frac{\mathbf{S}_{\mathbf{q}}^{\mu}}{\beta_\mu}$, and $\{\beta_\mu\}$ are sublattice dependent parameters.

Using Lagrange multipliers to incorporate the constraint in equation 15 gives the condition that the state which minimizes the energy must satisfy the eigenvalue equation

$$\sum_{\nu} L_{\mathbf{q}}^{\mu\nu} \mathbf{t}_{\mathbf{q}}^{\nu} = \lambda \mathbf{t}_{\mathbf{q}}^{\mu}, \tag{16}$$

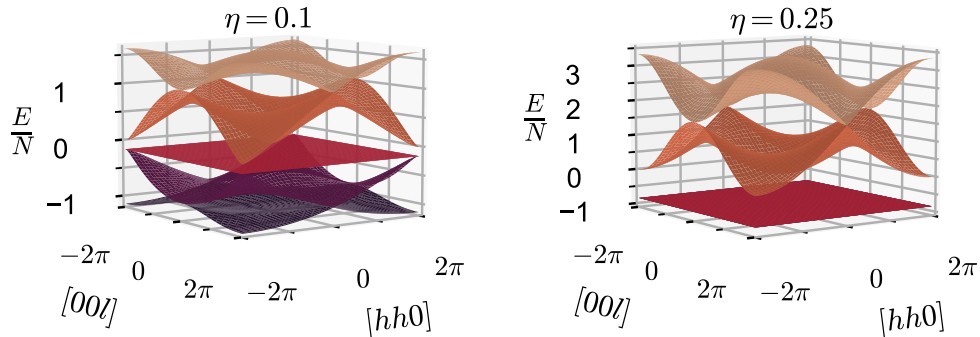

Figure 3: Energy spectrum obtained from the generalized LT method. For $\eta < 0.25$ (left), there is a unique ferrimagnetic ground state, corresponding to the band minimum at $\mathbf{q} = 0$. For $\eta > 0.25$ (right), the ground state is defined by a four-fold degenerate flat band with a gap to excitations.

with energy per unit cell

$$\epsilon = \lambda \sum_\mu \frac{1}{\beta_\mu^2}, \tag{17}$$

where the matrix $L_{\mathbf{q}}^{\mu\nu} = \beta_\mu \beta_\nu K_{\mathbf{q}}^{\mu\nu}$. As before, a candidate ground state can be found by placing all of the weight into the mode corresponding to the minimum eigenvalue (over all $\mathbf{q}$) of $L_{\mathbf{q}}^{\mu\nu}$. But now the eigenvalues and eigenvectors of $L_{\mathbf{q}}^{\mu\nu}$ depend on the $\{\beta_\mu\}$ so these can be tuned to ensure that the proposed ground state also satisfies equation 13.

For our model, we make the ansatz that

$$\beta_\mu = \begin{cases} 1, & \mu = a, b, \\ \beta, & \mu = 1, ..., 4, \end{cases} \tag{18}$$

which means the matrices in the standard and generalized variants of the LT are related by

$$L_{\mathbf{q}}^{\mu\nu}(\beta, \eta) = \beta K_{\mathbf{q}}^{\mu\nu}(\eta_{\text{eff}} = \beta \eta), \tag{19}$$

where $K_{\mathbf{q}}^{\mu\nu}$ is evaluated for a rescaled effective $\eta$, dependent on the $\beta$ we choose. For $0 < \eta < \frac{1}{4}$, we recover the known ferrimagnetic ground state by setting $\beta = \sqrt{\frac{2}{1-3\eta}}$. On the other hand, for $\eta \geq \frac{1}{4}$, an important observation is that at $\eta_{\text{eff}} = \frac{1}{\sqrt{2}}$ the spectrum of $K_{\mathbf{q}}^{\mu\nu}$ consists of a lower four-fold degenerate flat band and two higher dispersive bands. This degeneracy can be preserved in the spectrum of $L_{\mathbf{q}}^{\mu\nu}$ for arbitrary $\eta$ by choosing $\beta = \frac{1}{\sqrt{2}\eta}$, ensuring

$$L_{\mathbf{q}}^{\mu\nu}(\beta, \eta) \propto K_{\mathbf{q}}^{\mu\nu}\left(\eta_{\text{eff}} = \frac{1}{\sqrt{2}}\right). \tag{20}$$

From eq. 17 one obtains the energy corresponding to the minimum eigenvalues

$$\frac{E}{J_1 N} = -\frac{1}{6\eta} - \frac{2\eta}{3}. \tag{21}$$

Comparing to eq. 4, we know that this is the ground state energy of the system for $\eta \geq \frac{1}{4}$. Therefore, assuming that equation 13 can be satisfied by forming superpositions of the flat

band modes, we have found physical ground states of the system. Note that in this construction $\beta$ is continuous across the boundary at $\eta = \frac{1}{4}$.

To summarize, for $\eta \geq \frac{1}{4}$, the CPH ground state may be described in terms of a four-fold degenerate flat band. The full energy spectrum obtained using the generalized LT method is displayed in figure 3. Besides the increased number of flat bands, there is a gap in the mean-field spectrum, whereas for the kagome [38] and pyrochlore [39] the spectrum is gapless.

### 4.3.2 Connectivity matrix

Here, we reiterate the application of the method from refs. [42, 43] to the centred pyrochlore lattice, originally presented in [1], as it provides complementary evidence the ground state corresponds to a four-fold degenerate flat band for $\eta \geq \frac{1}{4}$. The Hamiltonian in the form of equation 4, can be rewritten in terms of an $\frac{N}{3} \times N$ connectivity matrix, $A_{t,n}$,

$$H = \frac{J_2}{2} \sum_{t=1}^{N/3} \sum_{n,m=1}^{N} A_{t,n} A_{t,m} \mathbf{S}_n \cdot \mathbf{S}_m \,, \tag{22}$$

where the constant term has been dropped. The elements of A are given by

$$A_{t,n} = \begin{cases} 1\,, & \text{if } n \in \text{vertices of } t\,, \\ \gamma\,, & \text{if } n \in \text{centre of } t\,, \\ 0\,, & \text{otherwise.} \end{cases} \tag{23}$$

The labels $n, m$ enumerate all sites of the lattice, whereas $t$ enumerates the tetrahedra. The dimension of the null space of A imposes a limit on the number of zero modes of $H$ and thus on the number of flat bands. For $\eta \geq \frac{1}{4}$ the minimum energy of $H$ as written in eq. 22 is zero, so these zero modes make up the ground state. Since

$$\text{rank(A)} \leq \frac{N}{3}\,, \tag{24}$$

the dimension of the null space,

$$\text{Nullity(A)} \geq N - \frac{N}{3} = \frac{2N}{3}\,, \tag{25}$$

by the rank-nullity theorem [44, 45]. The dimension of a band in momentum space is $\frac{N}{6}$, so 4 out of 6 bands of the mean-field energy spectrum of the CPHAF must belong to the ground state.

## 5 Phase diagram

Moving beyond mean-field methods, we obtain the finite temperature phase diagram in figure 1b from MC simulations. In particular we identify distinct regimes of what (on the mean-field level) is expected to be the disordered region of the model. Some important thermodynamic quantities as calculated from MC simulations for various $\eta$ are displayed in figs. 4a-c. Definitions for quantities computed in MC are given in appendix A. We also define a ferrimagnetic order parameter,

$$f = \langle \mathbf{m}_{\text{centres}} \cdot \mathbf{m}_{\text{vertices}} \rangle\,, \tag{26}$$

which is $-1$ in the saturated ferrimagnet and 0 in a paramagnet. The various low temperature phases, which we define by their features in the $T \to 0$ limit, are described as follows.



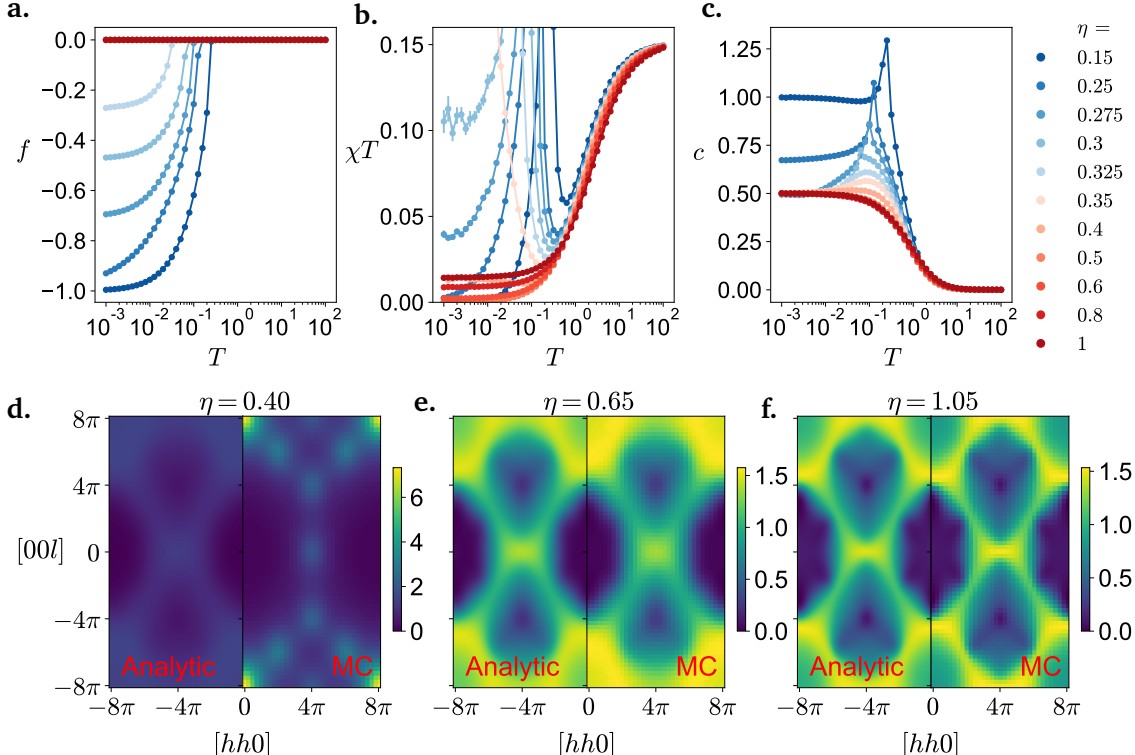

Figure 4: **a-c.** MC results of bulk thermodynamic quantities for various $\eta$. **a.** The ferrimagnetic order parameter (eq. 26). In the range $0.25 \leq \eta \leq 0.325$ its (finite) $T \to 0$ value decreases continuously, until vanishing in the SL phase. **b.** The susceptibility exhibits a low temperature Curie law, $\chi T = $ const for $\eta > \frac{1}{4}$. The low $T$ Curie constant decreases to zero at $\eta = 0.5$ before increasing again. **c.** The specific heat, $c(T \to 0) \to 0.5$ for all $\eta > \frac{1}{4}$, indicative of soft fluctuation modes about the ground state manifold. **d-f.** Structure factors calculated from mean-field (left panels) and MC at $T = 0.005$ (right panels). For $\eta < 0.4$ (**d.**), the mean-field calculation does not capture the broad maxima observed in MC. For $\eta > 0.5$ (**e,f**), the structure factor is characterized by broadened pinch points whose width decreases as $\eta$ is increased (see also [1]).

**Ferrimagnet,** $0 < \eta \leq \frac{1}{4}$:

The state identified analytically in section 4.1, with saturated ferrimagnetic order as $T \to 0$, magnetization $m_{\mathrm{all}} = \frac{1}{3}$ and $f = -1$. Low energy excitations about the ground state are transverse spin waves so the specific heat $c \to 1$ as $T \to 0$.

**Partial Ferrimagnet (PF),** $\frac{1}{4} < \eta \lesssim 0.343(3)$:

This phase is characterized by unsaturated ferrimagnetic order, $m_{all} < \frac{1}{3}$ and $f > -1$, with both continuously approaching zero as $\eta$ is increased. Fluctuations which preserve the local constraint, equation 6, are allowed, giving rise to zero modes which lower the heat capacity below 1 at the boundary ($\eta = 1/4$) and to $c = \frac{1}{2}$ for $\eta > 1/4$. We also observe a low temperature Curie law, $\chi T = $ const, usually a signature of a spin liquid [46], below the ordering transition. In the structure factor we do not observe any additional features beyond those associated with peaks at momenta corresponding to ferrimagnetic ordering. The coexistence of long-range order and fluctuations in the PF is superficially reminiscent of magnetic fragmentation in Coulomb spin liquids [21], however as we discuss in section 6 we do not expect an emergent field description to capture this.

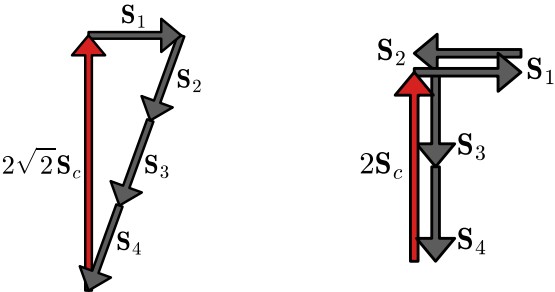

Figure 5: Diagrams representing the single tetrahedron spin configurations which satisfy eq. 6 for $\eta = \frac{1}{2\sqrt{2}}$ (left) and $\eta = \frac{1}{2}$ (right) whilst allowing for one or two spins respectively to be perpendicular to the central spin. Increasing $\eta$ decreases the effective central spin length, $\frac{1}{\eta}\mathbf{S}_c$, meaning that the perpendicular spins can acquire a finite component parallel to the direction of the centre spin, whereas decreasing $\eta$ leads to a finite antiparallel component for at least one of these spins. We propose this as a qualitative explanation for the different correlation regimes we observe in our MC simulations.

**Spin Liquid (SL), $\eta \gtrsim 0.343(3)$:**
We do not identify long range order in the magnetization or nematic order parameter, $Q^{(2)}$, nor do we find peaks in the specific heat or susceptibility. The susceptibility displays a Curie law crossover, where the low temperature Curie constant decreases continuously as $\eta$ increases, reaching zero at $\eta = 0.5$, before again increasing continuously with $\eta$. As in the partial ferrimagnet, $c = \frac{1}{2}$, indicative of the zero modes allowed by the local constraint. We can further distinguish two different regimes of the spin liquid by the spin structure factor. Firstly, for $0.343 \lesssim \eta \lesssim 0.5$, the structure factor is characterized by broad maxima at momenta associated with ferrimagnetic ordering (fig. 4d), indicative of short range ferrimagnetic correlations in the ground state. Secondly, for $\eta \gtrsim 0.5$, diffuse broadened pinch points are the key features of the structure factor (figs 4e,f). The width of these pinch points decreases as the pyrochlore limit, $\eta \to \infty$, is approached. These regimes of the structure factor evolve continuously into one another as $\eta$ crosses 0.5.

We can qualitatively rationalize the location of the different correlation regimes in parameter space by inspecting the single tetrahedron configurations allowed by the local constraint (eq. 6) in more detail. For $\frac{1}{4} \leq \eta \leq \frac{1}{2\sqrt{2}}$ all vertex spins must have a component anti-parallel to the central spin, as illustrated in fig. 5. Enforcing this on closed loops in the lattice would restrict the degree to which the central spins may deviate from pointing along a global direction, giving rise to long-range partial ferrimagnetic order. Then for $\eta > \frac{1}{2\sqrt{2}}$, a vertex spin may have a component parallel to the central spin. This would weaken the correlations between neighbouring central spins and could destroy any long-range order in the system. In MC simulations, extrapolating to the $L \to \infty$ limit at $T = 0.005$, the transition between the PF and SL occurs at $\eta = 0.343(3)$, not too far away from the predicted value of $\eta = \frac{1}{2\sqrt{2}} \approx 0.354$. A similar effect could be responsible for the change in correlations across $\eta = 0.5$, with 1 or 2 vertex spins allowed components parallel to the central spin for $\eta < 0.5$ and $\eta > 0.5$ respectively (see fig. 5).

# 6 Spin liquid

## 6.1 Mean-field structure factor

To calculate the ground state structure factor in the regime governed by the local constraint, we employ Henley's (approximate) projection-based approach [47]. This method is equivalent to the lowest order of a large-$N$ expansion (e.g ref. [9] on the pyrochlore) and was recently employed to distinguish classical spin liquids from a topological perspective [48].

We are interested in the regime where the ground state is defined by eq. 6, so restrict our attention to $\eta > \frac{1}{4}$. On the centred pyrochlore lattice, taking the Fourier transform of eq. 5 yields

$$\mathbf{L}_x(\mathbf{q}) = \gamma \mathbf{S}_{c_x}(\mathbf{q}) + \sum_{m=1}^{4} e^{\pm i \mathbf{q} \cdot \delta_m} \mathbf{S}_m(\mathbf{q}) = 0 \,, \tag{27}$$

where $x = a, b$ labels the tetrahedra centred on the corresponding sublattice, with spin $\mathbf{S}_{c_x}$ occupying the centre site. The exponent takes positive (negative) sign for $x = a(b)$ and the second equality is the ground state constraint. This may be rewritten in vector form as

$$\mathbf{L}_x(\mathbf{q}) = \vec{L}_x(\mathbf{q}) \cdot \vec{\mathbf{S}}(\mathbf{q}) = 0 \,, \tag{28}$$

where

$$
\begin{aligned}
\vec{L}_a(\mathbf{q}) &= (\gamma, 0, e^{i\mathbf{q}\cdot\delta_1}, e^{i\mathbf{q}\cdot\delta_2}, e^{i\mathbf{q}\cdot\delta_3}, e^{i\mathbf{q}\cdot\delta_4})^T \,, \\
\vec{L}_b(\mathbf{q}) &= (0, \gamma, e^{-i\mathbf{q}\cdot\delta_1}, e^{-i\mathbf{q}\cdot\delta_2}, e^{-i\mathbf{q}\cdot\delta_3}, e^{-i\mathbf{q}\cdot\delta_4})^T \,, \\
\vec{\mathbf{S}}(\mathbf{q}) &= (\mathbf{S}_{c_a}(\mathbf{q}), \mathbf{S}_{c_b}(\mathbf{q}), \mathbf{S}_1(\mathbf{q}), \mathbf{S}_2(\mathbf{q}), \mathbf{S}_3(\mathbf{q}), \mathbf{S}_4(\mathbf{q}))^T \,.
\end{aligned} \tag{29}
$$

The key object is the $6 \times 2$ matrix

$$\mathrm{E} = \begin{pmatrix} \vec{L}_a^* & \vec{L}_b^* \end{pmatrix} \,, \tag{30}$$

whose columns are the $\vec{L}_x^*$. Assuming weakly interacting spins, such that the probability distribution of spin configurations is Gaussian in the spin variables and enforcing equation 28 by projecting onto the subspace orthogonal to the $L_x^*$, the structure factor is given by

$$\langle \mathbf{S}_\mu(-\mathbf{q}) \cdot \mathbf{S}_\nu(\mathbf{q}) \rangle = s_0^2 \mathrm{P}_{\mu\nu}(\mathbf{q}) \,, \tag{31}$$

where $\mu, \nu$ label the sublattices, $s_0^2$ is a normalization constant and

$$\mathrm{P}_{\mu\nu}(\mathbf{q}) = \delta_{\mu\nu} - [\mathrm{E}(\mathrm{E}^\dagger \mathrm{E})^{-1} \mathrm{E}^\dagger]_{\mu\nu} \,. \tag{32}$$

Enforcing spin normalization on average, the structure factor over all sublattices (see eq. A.4) is

$$S(\mathbf{q}) = \frac{N_{\mathrm{u.c}}}{N} \sum_{\mu,\nu} \langle \mathbf{S}_\mu(-\mathbf{q}) \cdot \mathbf{S}_\nu(\mathbf{q}) \rangle = \frac{1}{\mathrm{Tr}(\mathrm{P})} M^T \mathrm{P} M \,, \tag{33}$$

where $M = (1, 1, 1, 1, 1, 1)^T$.

Pinch point singularities may arise in the structure factor at the $\mathbf{q}$ where $\mathrm{E}^\dagger \mathrm{E}$ is singular. Since

$$\det(\mathrm{E}^\dagger \mathrm{E}) = (\gamma^2 + 4)^2 - \left| \sum_{m=1}^{4} e^{2i\mathbf{q}\cdot\delta_m} \right|^2 \,, \tag{34}$$

for any finite $\gamma$, $\det(\mathrm{E}^\dagger \mathrm{E}) \neq 0$ and thus we do not expect to find pinch point singularities in the structure factor on the centred pyrochlore lattice. This is confirmed by our MC simulations.

Since these mean-field structure factors are for $T = 0$, results are displayed alongside those from low $T$ MC simulations in figs. 4d-f. We find good agreement for $\eta > 0.5$ so therefore expect that a long wavelength effective description is appropriate in this regime. Although the structure factor here does not have sharp pinch points for any finite $\eta$, the finite width pinch points suggest a close connection to the 3D Coulomb phase on the pyrochlore, which we explore in more detail in the next section. On the other hand, for $0.25 < \eta < 0.5$, we find mean-field deviates from MC; it cannot properly capture the short-range ferrimagnetic correlations which result from microscopically satisfying the local constraint. Nevertheless at intermediate temperature $T \approx 0.5$, mean-field and MC are in good agreement for all relevant $\eta$, even in the $0.25 < \eta < 0.5$ regime, likely due to the large entropy of the long wavelength spin liquid. This crossover from long wavelength spin liquid to short-range ferrimagnetic correlations could also explain the bump in specific heat seen for these values of $\eta$ around $T \approx 0.1$ in fig. 4c, which indicates a loss in entropy.

## 6.2 Coulomb physics

### 6.2.1 Charge fluid description

Here, we first restate the mapping (initially proposed in ref. [9]) which allows one to describe the PHAF ground state as a Coulomb phase, then explain how the centred pyrochlore geometry modifies this picture. We pointed out the resulting charge fluid description in ref. [1], but here we explain in more detail.

On each tetrahedron, at position $\mathbf{R}_t$, we define the three-component vector field

$$\mathbf{E}^\alpha(\mathbf{R}_t) = \sum_{m=1}^{4} \hat{\mathbf{u}}_m S^\alpha(\mathbf{R}_t \pm \boldsymbol{\delta}_m), \tag{35}$$

where there is one copy for each of the $\alpha = x, y, z$ spin components and use the orientation $\hat{\mathbf{u}}_m = \frac{\boldsymbol{\delta}_m}{|\boldsymbol{\delta}_m|}$ which points from $a$ to $b$ tetrahedra. The ground state condition for the PHAF is eq. 5 with $\gamma = 0$, which after coarse-graining translates to

$$\nabla \cdot \mathbf{E}^\alpha = 0. \tag{36}$$

Assuming a Gaussian effective free energy within the ground state manifold, the structure factor

$$E_{\mu\nu}^\alpha(\mathbf{q}) = \frac{1}{N_t} \sum_{t,t'} e^{i\mathbf{q}\cdot(\mathbf{R}_{t'}-\mathbf{R}_t)} \langle E_\mu^\alpha(\mathbf{R}_t) E_\nu^\alpha(\mathbf{R}_{t'}) \rangle, \tag{37}$$

where the $E_\mu^\alpha$ are the vector components of $\mathbf{E}^\alpha$ and the sum runs over all tetrahedra, $t, t'$, will take the form

$$E_{\mu\nu}^\alpha(\mathbf{q}) \propto \delta_{\mu\nu} - \frac{q_\mu q_\nu}{q^2}, \tag{38}$$

giving rise to characteristic pinch points at the centre of the Brillouin zone. In real space this corresponds to algebraic $1/r^3$ decay of correlations. The effective low energy theory is classical electrostatics, where excitations above the ground state introduce charges which interact via an (entropic in origin) Coulomb potential.

Now consider switching on a small, but finite, $\gamma$ in the local constraint on only $n$ 'defect' tetrahedra, whilst maintaining $\gamma = 0$ on all others. Provided these defects are well separated, after coarse graining the central spins on the defects can be viewed as $n$ charges

$$\nabla \cdot \mathbf{E}^\alpha(\mathbf{R}_t) = Q^\alpha(\mathbf{R}_t) \propto \pm \gamma S^\alpha(\mathbf{R}_t), \tag{39}$$

in each of the $\alpha$ channels with $-(+)$ on $a(b)$ tetrahedra. $Q^\alpha \in [-\gamma, \gamma]$ and therefore $\gamma$ parametrizes the maximum charge strength. Now the low-energy picture is that of three copies

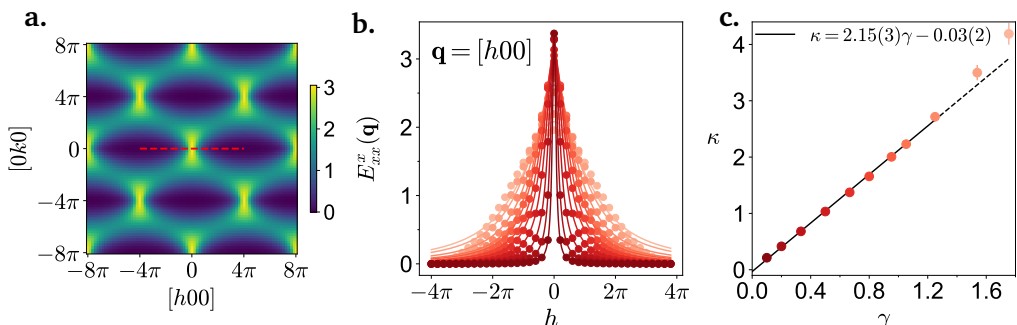

Figure 6: **a-c.** Computing the width of pinch points in $E_{xx}^x(\mathbf{q})$ from MC simulations, reproduced from [1]. **a.** The $E_{xx}^x(\mathbf{q})$ structure factor, eq. 37, as computed from MC for $\gamma = 0.67, T = 0.005$ in the $[hk0]$ plane. A cut is taken along the red line shown. **b.** Fitting the Lorentzian in eq. 41 to the MC data for various $\gamma$ (the same colours are used in **b** and **c**). **c.** $\kappa$ which parametrizes the width of the pinch point against $\gamma$, with linear fit up to $\gamma = 1.25$. The linear relation is characteristic of a dilute charge fluid with charge strength parameterized by $\gamma$.

of an emergent $U(1)$ gauge field, coupled to scalar charges on diamond lattice sites. Following the same arguments for the PHAF, these charges will experience an entropic effective Coulomb interaction. Then arguments from Debye-Hueckel theory [49–51] tell us that the field correlations must be screened as $e^{-\kappa r}$ with $\kappa \propto \gamma$, as any charge in the system carries a factor of $\gamma$. In momentum space, this results in the pinch points acquiring a finite width parametrized by $\kappa$

$$E_{\mu\nu}^\alpha(\mathbf{q}) \propto \delta_{\mu\nu} - \frac{q_\mu q_\nu}{q^2 + \kappa^2} \,. \tag{40}$$

Remarkably when we compute the structure factor of the CPHAF in MC simulations and fit to the form

$$E_{xx}^x(q_x, q_y = 0, q_z = 0) = \frac{A}{q_x^2 + \kappa^2} \,, \tag{41}$$

wth $A$ and $\kappa$ fitting parameters, we find that $\kappa \propto \gamma$ over a large region of the parameter space, $0 < \gamma \lesssim 1.25$. These results are summarized in fig. 6. This is despite the fact that the centred pyrochlore lattice corresponds to taking the limit $n \to N_t$ and Debye-Hueckel theory is used to describe systems of *dilute* charges at high temperature. Here there are charges, albeit with strength parametrized by $\gamma$, in at least one $\alpha$ channel on every tetrahedron (the effective temperature is a priori not known). The ground state can thus be viewed as the Heisenberg model variant of a monopole fluid in spin ice, for example studied in refs. [52,53]. This description does not impose any energetic constraints on the distribution of central spins, only accounting for how the central spins entropically rearrange themselves according to the effective electrostatic interactions between them. For small $\gamma$, we expect that all possible configurations of central spins will be allowed in the ground state. However for larger $\gamma$, certain configurations may no longer be energetically feasible and thus this view of the central spins as mobile charges will break down.

### 6.2.2 Analogy with pyrochlore thin films

Finite width pinch points have also been observed in the study of spin-ice thin-films [32], also featuring the existence of $\mathbb{Z}_2$ and $U(1)$ classical spin liquids, which we found for the Ising model on the centred pyrochlore. The connection between pyrochlore thin-films and the centred pyrochlore lattice can be clarified by mapping the centred pyrochlore to a slab of a 4D lattice of corner-sharing pentachora, which we term the pentachore lattice.

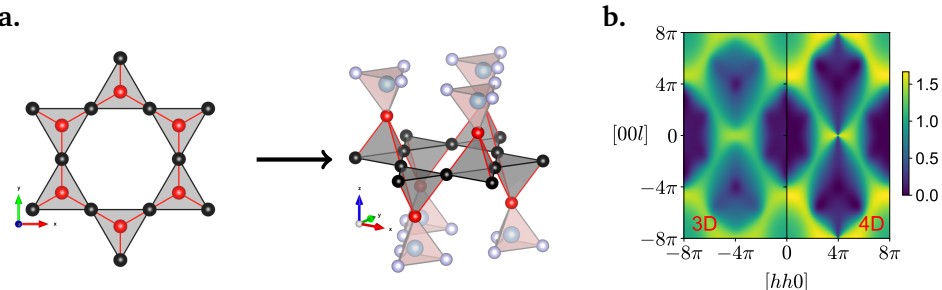

Figure 7: **a.** Mapping of the 2D centred kagome lattice to a slab of the pyrochlore lattice in 3D, analogous to the mapping of the 3D centred pyrochlore lattice to a slab of the 4D pentachore lattice. Left: The centred kagome lattice made up of corner-sharing centred triangles. Right: The corresponding slab of the pyrochlore lattice. Bulk tetrahedra are grey, whereas the pink virtual tetrahedra above and below the slab host unordered surface charges (blue) in the spin liquid ground state. **b.** Structure factor, $S(\mathbf{q})$, in the $[hhl0]$ plane calculated using the analytical mean-field calculation for $\eta = 1$ for the 3D centred pyrochlore lattice (left) and 4D pentachore lattice (right). The sharp pinch points on the 4D lattice along $[4\pi, 4\pi, l, 0]$ become broadened in the 3D case as pinch point singularities are not allowed by the symmetry of the 3D lattice.

The slab geometry is obtained by shifting central sites of the $a(b)$ tetrahedra alternately by $\delta_t = +(-)\frac{\sqrt{5}}{8}$, whilst the vertex spins remain in the $t = 0$ hyperplane, where $t$ is the additional Cartesian coordinate needed to describe the 4D space. Thus the slab has open boundaries at the $t = \pm\frac{\sqrt{5}}{8}$ edges. To illustrate the idea, the analogous 2D to 3D mapping from a centred kagome lattice to a slab of the pyrochlore lattice is shown in fig. 7a. We study this analogous situation in more detail in the next section. Returning to 4D, the slab can be generalized to a fully periodic pentachore lattice, specified by the positions

$$\mathbf{r}_{I,\mu}^{(4)} = \mathbf{R}_I^{(4)} + \boldsymbol{\delta}_\mu^{(4)}, \tag{42}$$

with lattice vectors

$$\mathbf{R}_I^{(4)} = n_1\mathbf{a}_1 + n_2\mathbf{a}_2 + n_3\mathbf{a}_3 + n_4\mathbf{a}_4, \tag{43}$$

where,

$$\mathbf{a}_1 = \frac{1}{2}\begin{pmatrix} 1 \\ 1 \\ 0 \\ 0 \end{pmatrix}, \qquad \mathbf{a}_2 = \frac{1}{2}\begin{pmatrix} 1 \\ 0 \\ 1 \\ 0 \end{pmatrix}, \qquad \mathbf{a}_3 = \frac{1}{2}\begin{pmatrix} 0 \\ 1 \\ 1 \\ 0 \end{pmatrix}, \qquad \mathbf{a}_4 = \frac{1}{4}\begin{pmatrix} -1 \\ -1 \\ -1 \\ \sqrt{5} \end{pmatrix}, \tag{44}$$

and basis vectors

$$\boldsymbol{\delta}_1^{(4)} = \frac{1}{8}\begin{pmatrix} 1 \\ 1 \\ 1 \\ -\frac{1}{\sqrt{5}} \end{pmatrix}, \qquad \boldsymbol{\delta}_2^{(4)} = \frac{1}{8}\begin{pmatrix} -1 \\ -1 \\ 1 \\ -\frac{1}{\sqrt{5}} \end{pmatrix}, \qquad \boldsymbol{\delta}_3^{(4)} = \frac{1}{8}\begin{pmatrix} 1 \\ -1 \\ -1 \\ -\frac{1}{\sqrt{5}} \end{pmatrix},$$

$$\boldsymbol{\delta}_4^{(4)} = \frac{1}{8}\begin{pmatrix} -1 \\ 1 \\ -1 \\ -\frac{1}{\sqrt{5}} \end{pmatrix}, \qquad \boldsymbol{\delta}_c^{(4)} = \frac{1}{8}\begin{pmatrix} 0 \\ 0 \\ 0 \\ \frac{4}{\sqrt{5}} \end{pmatrix}. \tag{45}$$

Note that the pentachore lattice has a 5 site basis, as the sites which in the slab geometry can be identified with central sites of the centred pyrochlore lattice become equivalent in the translational sense and are shared between neighbouring pentachora.

What can we say about the ground state on the periodic pentachore lattice? The generalization of our model will have a ground state (for $\eta > \frac{1}{4}$) defined by an analogous local constraint to equations 5 and 6, on each pentachoron. Crucially, now *all* spins are shared between clusters centred on a bipartite lattice. We can modify the mean-field structure factor calculation to account for the pentachore lattice geometry by letting

$$\vec{L}_a(\mathbf{q}) = (\gamma e^{i\mathbf{q}\cdot\delta_c^{(4)}}, e^{i\mathbf{q}\cdot\delta_1^{(4)}}, e^{i\mathbf{q}\cdot\delta_2^{(4)}}, e^{i\mathbf{q}\cdot\delta_3^{(4)}}, e^{i\mathbf{q}\cdot\delta_4^{(4)}})^T,$$
$$\vec{L}_b(\mathbf{q}) = (\vec{L}_a(\mathbf{q}))^*, \tag{46}$$
$$\vec{S}(\mathbf{q}) = (S_c(\mathbf{q}), S_1(\mathbf{q}), S_2(\mathbf{q}), S_3(\mathbf{q}), S_4(\mathbf{q}))^T,$$

where $(\dots)^*$ is the element-wise complex conjugate and redefining the E matrix accordingly. Now

$$\det(E^\dagger E) = (\gamma^2 + 4)^2 - \left| \gamma^2 e^{2i\mathbf{q}\cdot\delta_c^{(4)}} \sum_{m=1}^{4} e^{2i\mathbf{q}\cdot\delta_m^{(4)}} \right|^2, \tag{47}$$

which vanishes at certain $\mathbf{q}$, where the structure factor may exhibit pinch point singularities. Therefore we see that the symmetry of the centred pyrochlore lattice (equivalent to a pentachore slab) does not allow for sharp pinch points in the structure factor, while they are present for the fully periodic 4D pentachore lattice. Plots of the structure factor in both cases are shown in fig. 7b.

The sharp pinch points on the pentachore lattice suggest that there is a description of the ground state in terms of a 4D Coulomb phase. Indeed, one can define such a Coulomb phase in terms of a four-component vector field which is the higher dimensional version of $\mathbf{E}^\alpha$ (eq. 35). We introduce an orientation for the field with the unit vectors

$$\hat{\mathbf{u}}_\mu^{(4)} = 2\sqrt{5}\,\delta_\mu^{(4)}, \qquad \mu = 1, 2, 3, 4, c, \tag{48}$$

and then define the four-component vector field

$$\mathbf{E}^\alpha(\mathbf{R}_p^{(4)}) = \gamma \hat{\mathbf{u}}_c^{(4)} S^\alpha(\mathbf{R}_p^{(4)} \pm \delta_c^{(4)}) + \sum_{m=1}^{4} \hat{\mathbf{u}}_m^{(4)} S^\alpha(\mathbf{R}_p^{(4)} \pm \delta_m^{(4)}), \tag{49}$$

at the centre of each pentachoron, $\mathbf{R}_p^{(4)}$, for each spin component, $\alpha = x, y, z$ with $\pm = +(-)$ for $a(b)$ pentachora. Note that the $E_x^\alpha, E_y^\alpha, E_z^\alpha$ vector components of the $\mathbf{E}^\alpha = (E_x^\alpha, E_y^\alpha, E_z^\alpha, E_t^\alpha)$ field are proportional to the corresponding components of the 3D field. After coarse-graining, the ground state constraint, eq. 6, becomes

$$\text{div}(\mathbf{E}^\alpha(\mathbf{r}^{(4)})) = 0. \tag{50}$$

Following the same arguments as for the pyrochlore we expect sharp pinch points in the structure factor and a $1/r^4$ decay of correlations in real space. Defects would interact via a $1/r^2$ effective Coulomb potential; the effective theory for dilute defects is 4D electrostatics.

Returning to the pentachore slab, this can be viewed as the thinnest possible thin-film geometry which keeps both $a$ and $b$ pentachora of the lattice intact. Therefore, we can understand the properties of the ground state of the Heisenberg model on the centred pyrochlore lattice in a similar way to the spin-ice thin films studied by Lantagne-Hurtubise, Rau and Gingras (L-HRG) in ref. [32]. There, the authors considered various geometries of thin films of nearest neighbour spin-ice, also including so-called orphan bonds; bonds at the surface which

do not belong to a bulk tetrahedron, but instead can be thought of as belonging to a fictitious virtual tetrahedron. In our case, the central spins lie on the surface in the slab geometry; they not only belong to a bulk pentachoron but are also the single spin of a virtual pentachoron, see fig. 7a. As a result, the closest analogue to the pentachore slab (albeit in 3D rather than 4D) studied in ref. [32] is the [001] thin-film with orphan bonds set to zero. For such systems, L-HRG showed that at low $T$ the structure factor will also be characterized by finite width pinch points, which they argue is the result of fluctuating surface charges on the virtual tetrahedra. In our model, the flux entering/exiting a virtual pentachoron is $\gamma S_c^\alpha$, so is a continuous variable in $[-\gamma, \gamma]$. Note that due to the spin length constraint on a Heisenberg spin, there must be non-zero flux entering each bulk pentachoron in at least one of the $\alpha = x, y, z$ channels. Therefore we would also expect these fluctuating surface charges to destroy the Coulomb phase, with a screening length proportional to $\gamma$, as seen in our MC simulations.

The descriptions in terms of surface charges in higher dimensions or bulk charges as presented in the previous section describe the same effect. On the microscopic level, a spin at a surface or centre belongs only to a single (bulk) unit and therefore is less constrained than a spin shared between two corner-sharing units, modifying the allowed spin correlations in the ground state manifold in such a way as to destroy the Coulomb phase. The analogy between thin films and centred lattices can be useful in considering how to induce a desired effect in either system through the addition of perturbations as well as giving insight into the physics described by the bare Hamiltonian.

## 7 Centred kagome lattice

The mechanisms discussed on the centred pyrochlore lattice, whereby the central spins act as mobile sources of flux, should also apply to other lattices made up of suitable corner-sharing centred clusters, regardless of dimensionality or number of spins making up the cluster. Therefore, we also investigate the Hamiltonian, eq. 3, on the 2D analogue of the centred pyrochlore lattice, the centred kagome lattice. Here, an additional site at the centre of each triangular unit of the kagome lattice is coupled to the vertex spins by $J_1$ and the vertex spins are mutually coupled by $J_2$.

The centred kagome lattice is defined by the position vectors

$$\mathbf{r}_{i,\mu}^{(2)} = \mathbf{R}_i^{(2)} + \boldsymbol{\delta}_\mu^{(2)}, \tag{51}$$

where $\mathbf{R}_i^{(2)} = n_1 \mathbf{a}_1^{(2)} + n_2 \mathbf{a}_2^{(2)}$, with the triangular lattice vectors $\mathbf{a}_1^{(2)} = \frac{1}{2}(1, -\sqrt{3})$, $\mathbf{a}_2^{(2)} = \frac{1}{2}(-1, -\sqrt{3})$ and integer $n_1, n_2$. We choose units of $\left|\mathbf{a}_1^{(2)}\right| = \left|\mathbf{a}_2^{(2)}\right| = 1$. The lattice has a 5-site basis of

$$\boldsymbol{\delta}_a^{(2)} = \begin{pmatrix} 0 \\ 0 \end{pmatrix}, \qquad \boldsymbol{\delta}_b^{(2)} = \begin{pmatrix} 0 \\ \frac{1}{\sqrt{3}} \end{pmatrix}, \tag{52}$$

$$\boldsymbol{\delta}_1^{(2)} = \begin{pmatrix} 0 \\ \frac{1}{2\sqrt{3}} \end{pmatrix}, \qquad \boldsymbol{\delta}_2^{(2)} = \begin{pmatrix} \frac{1}{4} \\ \frac{-1}{4\sqrt{3}} \end{pmatrix}, \qquad \boldsymbol{\delta}_3^{(2)} = \begin{pmatrix} \frac{-1}{4} \\ \frac{-1}{4\sqrt{3}} \end{pmatrix}.$$

Sites labelled by $a(b)$ occupy the centre of up (down) triangles. As for the centred pyrochlore, the Hamiltonian may be rewritten in the form of eq. 4, which gives rise to the ground state constraint

$$L_t = 0, \qquad \forall t, \tag{53}$$

for $\eta \geq \frac{1}{3}$, where

$$\mathbf{L}_t = \gamma \mathbf{S}_{t,c} + \sum_{v=1}^{3} \mathbf{S}_{t,v}, \tag{54}$$

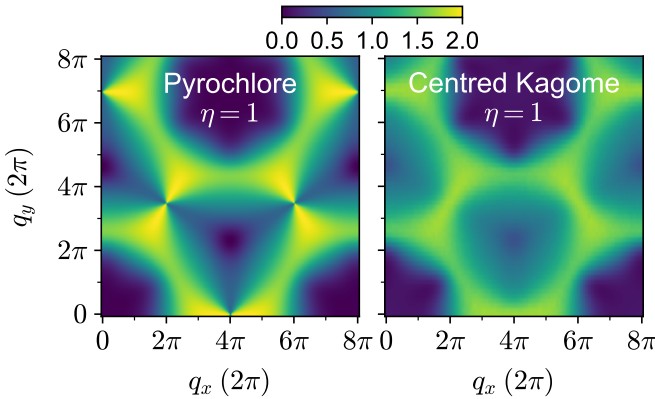

Figure 8: Comparison of the structure factor of the nearest neighbour Heisenberg model on the pyrochlore lattice (left, computed using the mean-field calculation in [47]) and the centred kagome lattice (right, computed from MC at $T = 0.02$ for a system size of $L = 48$). The sharp pinch points on the pyrochlore become broadened upon reducing the dimension to the centred kagome lattice.

with $t$ now labelling centred triangular units. For $\eta \leq \frac{1}{3}$, the energy is minimized by the ferrimagnetic state.

Analogous to the mapping from the 3D centred pyrochlore lattice to the 4D pentachore slab, one can map the 2D centred kagome lattice to a slab of the 3D pyrochlore lattice, as shown in fig. 7a. This is done by considering the centred kagome lattice as occupying the $z = 0$ plane of a 3D space, then shifting the central sites alternately up(down) to $z = \pm \frac{1}{\sqrt{6}}$, with $\pm = +(-)$ respectively. Thus the centred triangles become tetrahedra and we have a pyrochlore slab with open boundaries at the $z = \pm \frac{1}{\sqrt{6}}$ edges, which is the thinnest possible thin-film geometry keeping both $a$ and $b$ tetrahedra intact. Adapting the calculations of sec. 6.1 to the centred kagome it can be shown that the disordered state would not have any pinch point singularities for finite $\gamma$, as we would expect from the arguments of the previous sections. This is confirmed by our MC simulations on the centred kagome lattice where we find broadened pinch points in the structure factor, as shown in figure 8. Thus, we see that the picture of central spins acting as mobile sources of flux in the effective low energy field theory is not unique to the centred pyrochlore lattice.

# 8 $J_1 - J_2 - J_3$ model

Returning to the centred pyrochlore lattice, we now investigate the possibility to realize different states of matter by applying perturbations to the bare $J_1 - J_2$ Hamiltonian, targeting specific regions of the degenerate ground state manifold. In particular, we consider the effect of a $J_3$ term coupling next next nearest neighbours, i.e centre spins on adjacent tetrahedra.

## 8.1 Ferromagnetic $J_3$

After the addition of a ferromagnetic $J_3$ the ground state manifold is made up of states with ferromagnetic centre spins and vertex spins correspondingly satisfying the local constraint. Selecting the $\hat{\mathbf{z}}$ direction as that along which the centres are aligned, the local constraint can

be rewritten as

$$\sum_{\nu=1}^{4} \begin{pmatrix} S_\nu^x \\ S_\nu^y \\ S_\nu^z + \frac{\gamma}{4} \end{pmatrix} = \sum_{\nu=1}^{4} \tilde{\mathbf{S}}_\nu = 0 \,, \tag{55}$$

such that the rescaled vertex spins, $\tilde{\mathbf{S}}_\nu$ can be mapped to the usual divergence-less field of the 3D Coulomb phase (eq. 35). The structure factor of $\tilde{\mathbf{S}}_\nu$ should then yield sharp pinch points as well as a $1/r^3$ algebraic decay in real space. This is verified in MC simulations, by calculating the structure factor

$$S^\perp(\mathbf{q}) = \frac{1}{N} \sum_{i,j} \mathbf{S}_i^\perp \cdot \mathbf{S}_j^\perp e^{i\mathbf{q}\cdot(\mathbf{r}_i - \mathbf{r}_j)} \,, \tag{56}$$

where $\mathbf{S}_i^\perp = (S_i^x, S_i^y)^T$ and the orientation of the axes is chosen such that $\hat{\mathbf{z}} = \hat{\mathbf{m}}_{\text{centres}}$ for each spin configuration sampled. We find sharp pinch points for all $\eta \geq 0.3$ simulated. The full structure factor, eq. A.4, for $\eta = 0.4$ is presented in figure 9a, showing the coexistence of Bragg peaks and pinch points. Considering the local constraint, we would expect sharp pinch points to persist all the way down to $\eta = 1/4$, but the lower temperature required to enter the Coulomb phase at low $\eta$ and the reduced weight in the perpendicular spin components, makes their observation challenging as this limit is approached. Furthermore, we also calculate the real space spin correlations for perpendicular spin components and find the characteristic $1/r^3$ decay expected for a 3D Coulomb phase, as shown in figure 9b.

This recovery of the Coulomb phase can be easily understood in the effective field theory picture. When the central spins order ferromagnetically they form a perfect zinc blende charge crystal in the $z$ channel with charges $Q^z = \pm\gamma$ on alternating diamond sublattices. This leaves the $x$ and $y$ channels of the effective field with no charges, therefore restoring the divergence-free condition of the 3D Coulomb phase. This is analogous to the situation in spin ice thin films where one can stabilize the 2D Coulomb phase by inducing ordering in the surface charges [32, 54].

## 8.2 Antiferromagnetic $J_3$

Taking antiferromagnetic $J_3$ introduces additional frustration into the model as $J_1$ and $J_3$ bonds cannot be simultaneously satisfised; $J_1$ bonds connecting a pair of centre sites through an intermediate vertex spin favours ferromagnetic order of the centres, whereas $J_3$ favours Néel order. For $J_2 = 0$ one can think of the lattice as a singly decorated diamond lattice, where the basic frustrated unit can be represented as the triangle in fig. 9c. Minimizing the energy on a single triangle, the ground state is the same ferrimagnetic state as in the $J_1 - J_2$ model for $\frac{J_3}{J_1} \leq \frac{1}{2}$, whereas for $\frac{J_3}{J_1} > \frac{1}{2}$ the ground state is the canted state shown (fig. 9c), with the angle between the centres and their shared vertex spin $\cos\phi = -\frac{J_1}{2J_3}$. In the limit $\frac{J_3}{J_1} \to \infty$ this becomes a state with Néel ordered centre sites which are decoupled from the vertex sites.

On the full lattice, the ferrimagnetic state remains the ground state, however for the canted state the spiral order of the centre spins must be commensurate with closed loops on the lattice to guarantee it remains a ground state. This requires that $N_{loop}\phi = \pi n$ where $N_{loop}$ is the number of centre sites in a given loop. For example, considering only the shortest hexagonal loops, a commensurate spiral order of the centre sites is obtained at $\phi = \frac{2\pi}{3}$ and $\frac{5\pi}{6}$, corresponding to $\frac{J_3}{J_1} = 1$ and $\frac{J_3}{J_1} = \frac{1}{\sqrt{3}}$ respectively. The difference in energy between the Neel $\frac{J_3}{J_1} \to \infty$ ground state and the canted state is $E_N - E_c = \frac{J_1^2}{J_3}$. Therefore for large but finite $J_3$ the energy difference is small. Combined with the fact that commensurate spiral orders will not be possible on hexagonal loops for large $\frac{J_3}{J_1}$, this leaves the state with Neel ordered centres as the likely ground state.

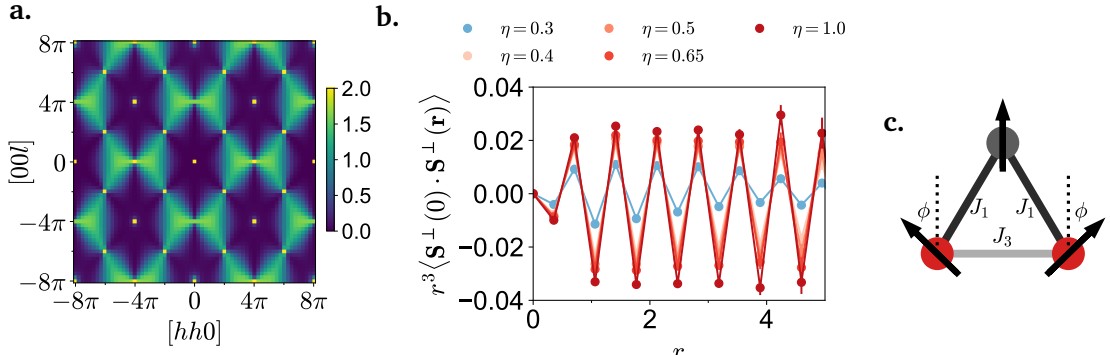

Figure 9: **a.** Structure factor for $\eta = 0.4$, $J_3 = -0.1$ at $T = 0.001$ computed from MC. The colour scale is restricted to a maximum of 2 to show the coexistence of Bragg peaks from long-range ordering of the centre spins and pinch points from the vertex spins. The width of the pinch points continues to decrease with temperature up to the resolution allowed by the finite system size. **b.** Real space spin correlations (for perpendicular spin components to the ordering axis) computed from MC in the [110] direction at $T = 0.001$. They decay as $1/r^3$ for all $\eta$ shown, except at $\eta = 0.3$, where we are not able to access low enough temperatures in our simulations. **c.** The basic frustrated unit of the antiferromagnetic $J_1 - J_3$ model on the centred pyrochlore lattice. For $\frac{J_3}{J_1} > \frac{1}{2}$ the ground state is that shown with the angle between centre spins (red) and their shared vertex spin (grey) given by $\cos \phi = -\frac{J_1}{2J_3}$.

Returning to the $J_1 - J_2 - J_3$ model and assuming Neel ordered centre sites, vertex sites become decoupled from central sites, so the lowest energy configuration is achieved when the vertices satisfy the usual pyrochlore local constraint, i.e eq. 6 with $\gamma = 0$. Therefore the vertex spins should realize the usual 3D Coulomb phase, which unlike in the ferro $J_3$ case does not require any rescaling of the spin. In MC simulations, we find a state with Neel ordered centres and vertex spins satisfying the local constraint with $\gamma = 0$ as $T \to 0$ (down to $T = 10^{-3}$) for a system size of $L = 4$ and $J_3 = 10$. However, upon increasing the system size, it becomes challenging to thermalize the MC simulations at low temperature. More experimentally relevant would be a small antiferromagnetic $J_3$, however MC simulations also struggle to thermalize in this case, so we were unable to identify which ground states such a perturbation would favour in the thermodynamic limit.

## 9  Summary and Outlook

The CPHAF hosts a highly degenerate spin liquid ground state over a large region of the parameter space. This gives rise to several unusual low temperature phases: a partial ferrimagnet where partial long-range order and fluctuations coexist, a disordered regime where the microscopic need to satisfy the ground state constraint leads to a short-range ferrimagnetically correlated ground state, and a spin liquid regime which admits an effective description in terms of a fluid of mobile charges. In the latter the central spins act as sources and sinks of flux of the effective field, interacting entropically via a Coulomb potential, causing screening of the spin correlations in the ground state and therefore broadening of pinch points in momentum space. This broadening of pinch points due to the addition of a central spin is not unique to the centred pyrochlore lattice; we find the same phenomenon on the 2D centred kagome lattice. One can connect this physics to that of thin films by viewing these $d$-dimensional lattices as

slabs in a higher $d + 1$ dimensional space, where the periodic $d + 1$ dimensional lattice would host a Coulomb phase ground state.

We have also shown how additional terms can be added to the pure $J_1 - J_2$ centred pyrochlore Hamiltonian in order to stabilize specific states of matter, in this case the 3D Coulomb phase with the addition of a ferromagnetic $J_3$. Therefore the centred pyrochlore lattice offers a new frustrated geometry to explore the rich physics of spin liquids.

Although the focus of this paper has been purely theoretical, we should remember that the centred pyrochlore lattice may be realized in the lab in (highly tunable) metal-organic frameworks [1]. Therefore with continued theoretical and experimental work there exists the real possibility to experimentally realize novel states of matter.

Going forward, there remain some open questions to fully characterize the low temperature states of the CPHAF. For $\eta \lesssim 0.8$ the effective description in terms of a dilute fluid of charges appears to break down; a correct description would probably have to incorporate the energetic constraints on central spin configurations, as well as corrections to the Debye-Hueckel theory arising from the fact that the charge density becomes large. Furthermore, we have not focussed much on the partial ferrimagnetic state, where a microscopic understanding of the system appears crucial. Understanding how to satisfy the local constraints on tetrahedra forming closed loops in the lattice is probably important, which may be possible to capture with an effective model on the diamond lattice. Returning to the regime characterized by broadened pinch points and in light of the $\mathbb{Z}_2$ nature of the Ising ground state over a broad range of $\eta$, future work could investigate whether the CPHAF realizes a $\mathbb{Z}_2$ classical Heisenberg spin liquid as introduced in ref. [55].

Further afield, it is interesting to consider whether quantum spin liquid ground states of quantum Hamiltonians exist on the centred pyrochlore lattice. For example, how does the addition of a central spin affect the $U(1)$ quantum spin liquid of the XXZ Hamiltonian on the pyrochlore? We have discussed how we expect the Ising model on the centred pyrochlore lattice to host both $\mathbb{Z}_2$ and $U(1)$ classical spin liquids, so could it be that the addition of quantum fluctuations would realize a $\mathbb{Z}_2$ quantum spin liquid?

On the experimental front, further experimental measurements such as NMR, $\mu$-SR or neutron scattering on Mn(ta)$_2$ would be useful to probe the putative proximate spin liquid above $\sim 1K$, to see if signatures of the spin liquid state can be observed, as well as verifying the nature of ordering below $T_c$. Away from the classical regime, the synthesis and measurement of Cu based MOFs realizing the centred pyrochlore lattice would be of great interest in order to probe the properties of $S = 1/2$ quantum spins in this geometry. Cu(ta)$_2$ has already been studied in ref. [56], where it was found that a Jahn-Teller distortion at low temperature breaks the cubic symmetry of the lattice and thus one would expect differing exchange interactions between the vertex spins of the tetrahedra. Nevertheless, one of the great strengths of metal-organic frameworks is their tunability, and it is possible that the correct choice of ligands may be able to preserve the cubic symmetry of the centred pyrochlore lattice down to low temperatures.

# Acknowledgements

We thank Han Yan, Richard Röß-Ohlenroth, Dirk Volkmer, Anton Jesche, Hans-Albrecht Krug von Nidda, Alexander Tsirlin and Phillip Gegenwart for valuable discussions and insights. Our simulations make use of the ALPSCore library [57].

**Funding information** R.P.N., L.P. and L.D.C.J. acknowledge financial support from the LMU-Bordeaux Research Cooperation Program. R.P.N. and L. P. acknowledge support from

FP7/ERC Consolidator Grant QSIM-CORR, No. 771891, and the Deutsche Forschungs-gemeinschaft (DFG, German Research Foundation) under Germany's Excellence Strategy –EXC-2111–390814868. L.D.C.J. acknowledges financial support from ANR-18-CE30-0011-01.

## A   Monte Carlo simulations

Classical Monte Carlo simulations on the centred pyrochlore lattice were performed for cubic systems of $N = 24L^3$ spins, where $L$ is the number of conventional unit cells along each Cartesian axis. On the centred kagome lattice systems of $N = 5L^2$ spins were used, where $L$ specifies the number of primitive unit cells spanning each direction of the triangular Bravais lattice. Each MC sweep consists of a sweep of overrelaxation [58–61] and heatbath [58,62,63] updates through the entire lattice.

Quantities computed during MC simulations include: magnetizations

$$\mathbf{m}_j = \frac{1}{N_j} \langle \sum_{i \in j} \mathbf{S}_i \rangle, \tag{A.1}$$

where $j$ may include all or a subset of spins on the lattice, the ferrimagnetic order parameter, eq. 26, the nematic order parameter, $Q^{(2)}$, defined in ref. [64], which measures quadrupolar moments, the magnetic susceptibility (per site)

$$\chi = \frac{N}{T} \left( \langle m_{\text{all}}^2 \rangle - \langle m_{\text{all}} \rangle^2 \right), \tag{A.2}$$

and specific heat (per site)

$$c = \frac{1}{T^2} \left( \langle E^2 \rangle - \langle E \rangle^2 \right), \tag{A.3}$$

where $E$ is the energy calculated according to equation 3.

To probe spin correlations we also compute the structure factor

$$S(\mathbf{q}) = \frac{1}{N} \sum_{i,j} e^{i\mathbf{q} \cdot (\mathbf{r}_j - \mathbf{r}_i)} \langle \mathbf{S}(\mathbf{r}_i) \cdot \mathbf{S}(\mathbf{r}_j) \rangle, \tag{A.4}$$

where $\mathbf{r}_i, \mathbf{r}_j$ are the position vectors of all lattice sites, enumerated by the indices $i, j$, as well as the structure factor of the effective field, defined in eq. 37. Simulations were performed up to system sizes of $L = 10$ when computing the structure factor and $L = 14$ for thermodynamic quantities.

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
