# Peer review of "The Classical Heisenberg Model on the Centred Pyrochlore Lattice"

_SciPost Physics, doi:SciPost Phys. 15, 040 (2023)_

## Round 2 · Referee Report · Anonymous (Referee 1) · 2023-4-6

Strengths

(1) Highly nontrivial connections are revealed which elucidate and enhance our understanding of the observed features.

(2) Addressing the long-standing problem of identifying parametrically enhanced stabilities of Coulomb spin liquids

(3) Opening new avenues for further explorations and bridging the gap between theory and experiment.

Weaknesses

Within the scope of what this work sets out to achieve, I do not find any major weaknesses as such.

Report

The manuscript “The Classical Heisenberg Model on the Centred Pyrochlore Lattice” by Nutakki et al. addresses the classical zero- and finite-temperature physics on a novel three-dimensional lattice dubbed the “Centred Pyrochlore Lattice” which is motivated from the material realization in the form of Mn(ta)2. Their phase diagram is shown to be host to an interesting array of phases and intriguing sequence of phase transitions and scaling behaviors. The structure of the ground state manifold is revealed to be more frustrated compared to the iconic pyrochlore lattice.

The highlight of the work is the finding of extended regimes of classical spin liquid behavior which interestingly finds a description in terms of a charged fluid discussion as highlighted by the authors. Another nontrivial aspect of the work is the rigorous connection that is established to the physics of thin films by viewing the 3d lattices as slabs of 4d parent lattices hosting a perfect Coulomb phase, and in particular, in the context of a previous important work on pyrochlore thin films. This proves crucial towards rationalizing many of the observations such as broadened structure factors instead of perfectly sharp ones found in 4d. This finding is likely to have wider ramifications towards furthering our understanding of other models in frustrated magnetism, and is an insight which adds significant value to the current manuscript. The authors also identify a direction in parameter space (addition of long-range Heisenberg ferromagnetic J3 coupling) which recovers the perfect Coulomb phase displaying singular pinch points, and this finds a nice explanation within a effective field theory picture. It is worth noting that given the highly delicate nature of such Coulomb spin liquid phases finding extended regions of their stability is a highly nontrivial task and constitutes an important finding.

In my opinion, this work opens many future avenues of exciting research, in particular, the impact of quantum fluctuations for small values of spin, e.g., S=1/2, and whether it can stabilize a quantum spin liquid. Given, that the classical T=0 ground state manifold is more under-constrained compared to the pyrochlore lattice, it is more or less expected to have a nonmagnetic ground state. The identification of "parent" classical models which upon introduction of quantum fluctuations are likely to host nonmagnetic ground states are of special significance. Given the authors insights on the Ising model, a microscopic wave function study probing for U(1) and Z2 spin liquid ground states as claimed for, is warranted. In light of the promising developments taking place in the experimental synthesis and realizations of metal-organic networks, this study is timely and contextual and will set the stage for future works both theoretical and connecting with experiment.

The manuscript is well written, with all details of calculations spelled out and scientific arguments being sound and conveyed in a logically coherent manner making them easy to follow. I thus recommend this manuscript for publication in SciPost.

Requested changes

None

---

## Round 2 · Referee Report · Anonymous (Referee 2) · 2023-5-20

Strengths

1- Theoretically, it establishes a field of research in a novel classical frustrated system that can be realized experimentally. This field promises to be very rich. The manuscript encourages further contributions from theory, simulations, and experiments. 2- The topics addressed, involving fake electromagnetism and topological matter in a classical context, could be of interest in a community exceeding that of classical frustrated magnetism. 3- The text is very clearly and carefully written. Some sections seem to be almost pedagogical, with an intention to raise the interest on the subject or connect with communities dedicated to different materials or systems (for example, sections 4.2, 6.2.2, 7).

Weaknesses

For sure any work can be improved or extended, but I would say the authors have made a good and careful job.

Report

The central result of this theoretical study on a classical Heisenberg model on a frustrated novel lattice, is the rich phase diagram shown in Fig. 1. Although the main parameter here (the ratio of two exchange constants) is generally far from being experimentally accesible, the versatility of the material (a family of metal-azolate frameworks) can make these results relevant for the planning and understanding of experiments on real samples.
Along the paper, the authors rationalize the existence and relative stability of these phases through a variety of analytical methods, and using computational simulations. The methods tend to be complementary, and help to construct a unified and convincing framework. The different classical topological spin liquids they find are results of particular interest. They are shown to have a high degree of degeneracy, and are described in terms of new degrees of freedom analogous to charges.
It is interesting, and provides a straight connection with other materials, that some of the characteristics of these phases can be explained or illustrated moving along the dimensionality axis. For example, the finite width of the pinch points in the magnetic structure factor is understood by looking at the system as a three-dimensional slab of a four dimensional lattice of corner-sharing pentachora. The argument is further developed by studying a centered kagome lattice as a slab of the three-dimensional pyrochlore one. Another resource, perhaps analogous to the previous one, is used to understand in a simpler way the phase diagram at low temperatures; here the Heisenberg spins are the ones that are projected into a one dimensional, Ising-like axis.
In the final subsections, the authors explain how to tune new physics from their simple Hamiltonian by including perturbations. In particular, they show how to stabilize a true Coulomb phase ground state using interactions up to third nearest neighbors.
The manuscript is very well written, and it may help other researchers (theoreticians or experimentalists, from similar and not so similar fields) to understand and be interested in the family of compounds they study here. Although this is a classical system, the authors indicate possible routes of research involving quantum spin liquid ground states.
Taking into account the points enumerated in the section dedicated to the strengths of the manuscript and all above, I will recommend this article to be published in SciPost.

Requested changes

I list below some minor points to noted and corrected. * Fig. 2e. and caption. The color convention for the spheres is not always evident. It may be worth mentioning there the red and dark spheres (the caption now only mentions the cyan and pink ones). * Flat bands. The existence of flat bands for \eta > 1/4 is established using the Luttinger-Tisza method, but only provided the “strong” constraint given by Eq. 13 can be satisfied. This provisional conclusion is made more robust in the following section of the manuscript (4.3.2), using the connectivity matrix method. I understand this is valid only on the same \eta range specified before; however, this restriction is not mentioned in the text (although, it is in the cited reference, [1] in the manuscript). * Page 12. Here the authors mention the value of the order parameter for the ferrimagnetic phase, f. However, I think this parameter is only defined in a following section, dealing with Monte Carlo simulations details (page 24).

---

## Round 3 · Author Response

We have revised the manuscript to address the comments of the second referee.

---

## Round 3 · List of Changes

- Changed the caption of fig. 2 to make the colour convention clear.
- Added explanation that the connectivity matrix is only valid for eta > 1/4 in section 4.3.2
- Moved the definition of the ferrimagnetic order parameter from the appendix to the beginning of section 5.

---

## Editorial Decision

published